# Clearance of senescent decidual cells by uterine natural killer cells in cycling human endometrium

**Paul J Brighton[1†], Yojiro Maruyama[1,2†], Katherine Fishwick[1], Pavle Vrljicak[1,3], Shreeya Tewary[1,3], Risa Fujihara[1,2], Joanne Muter[1], Emma S Lucas[1,3], Taihei Yamada[2], Laura Woods[4,5], Raffaella Lucciola[1], Yie Hou Lee[6,7], Satoru Takeda[2], Sascha Ott[3], Myriam Hemberger[4,5], Siobhan Quenby[1], Jan Joris Brosens[1]***

[1]Division of Biomedical Sciences, Clinical Science Research Laboratories, Warwick Medical School, University of Warwick, Coventry, United Kingdom; [2]Department of Obstetrics and Gynecology, Juntendo University Faculty of Medicine, Tokyo, Japan; [3]Tommy's National Centre for Miscarriage Research, University Hospitals Coventry and Warwickshire, Coventry, United Kingdom; [4]Centre for Trophoblast Research, University of Cambridge, Cambridge, United Kingdom; [5]Epigenetics Programme, The Babraham Institute, Cambridge, United Kingdom; [6]Obstetrics & Gynaecology Academic Clinical Program, Duke-NUS Medical School, Singapore, Singapore; [7]KK Research Centre, KK Women's and Children's Hospital, Singapore, Singapore

**\*For correspondence:**
J.J.Brosens@warwick.ac.uk

[†]These authors contributed equally to this work

**Competing interests:** The authors declare that no competing interests exist.

**Abstract** In cycling human endometrium, menstruation is followed by rapid estrogen-dependent growth. Upon ovulation, progesterone and rising cellular cAMP levels activate the transcription factor Forkhead box O1 (FOXO1) in endometrial stromal cells (EnSCs), leading to cell cycle exit and differentiation into decidual cells that control embryo implantation. Here we show that FOXO1 also causes acute senescence of a subpopulation of decidualizing EnSCs in an IL-8 dependent manner. Selective depletion or enrichment of this subpopulation revealed that decidual senescence drives the transient inflammatory response associated with endometrial receptivity. Further, senescent cells prevent differentiation of endometrial mesenchymal stem cells in decidualizing cultures. As the cycle progresses, IL-15 activated uterine natural killer (uNK) cells selectively target and clear senescent decidual cells through granule exocytosis. Our findings reveal that acute decidual senescence governs endometrial rejuvenation and remodeling at embryo implantation, and suggest a critical role for uNK cells in maintaining homeostasis in cycling endometrium.
DOI: https://doi.org/10.7554/eLife.31274.001

## Introduction

Different mammalian species employ divergent strategies to ensure successful embryo implantation. In mice, synchronized implantation of multiple embryos (average 6–8) is dependent on a transient rise in circulating estradiol (E2) that not only renders the progesterone-primed endometrium receptive, but also activates dormant blastocysts for implantation (*Paria et al., 1998*). Upon breaching of the uterine luminal epithelium, implanting murine embryos trigger extensive remodeling of the endometrial stromal compartment. This process, termed decidualization, is characterized by local edema, influx of uNK cells and differentiation of stromal fibroblasts into specialized decidual cells that coordinate trophoblast invasion and placenta formation (*Gellersen and Brosens, 2014; Zhang et al., 2013*). Likewise, the human endometrium transiently expresses a receptive phenotype,

lasting 2–4 days, during the mid-luteal phase of the cycle. However, this implantation window is not controlled by a nidatory E2 surge (*de Ziegler et al., 1992*; *Groll et al., 2009*), perhaps reflecting that synchronized implantation of multiple human embryos is neither required nor desirable. Further, decidualization of the stromal compartment is not dependent on an implanting embryo but initiated during the mid-luteal phase of each cycle in response to elevated circulating progesterone levels and increased intracellular cAMP production (*Gellersen and Brosens, 2014*). In parallel, CD56<sup>bright</sup> CD16<sup>−</sup> uNK cells accumulate in luteal phase endometrium. In pregnancy, uNK cells exert an evolutionarily conserved role in orchestrating vascular adaptation and trophoblast invasion (*Hanna et al., 2006*; *Xiong et al., 2013*), but their function in cycling human endometrium is unclear.

Differentiation of human endometrial stromal cells (EnSCs) into decidual cells is a multistep process (*Gellersen and Brosens, 2014*). Following cell cycle exit at G0/G1, decidualizing EnSCs first mount a transient pro-inflammatory response, characterized by a burst of free radical production and secretion of various chemokines and other inflammatory mediators (*Al-Sabbagh et al., 2011*; *Lucas et al., 2016b*; *Salker et al., 2012*). Exposure of the mouse uterus to this inflammatory secretome activates multiple receptivity genes, suggesting that the nidatory E2 surge in mice is supplanted by an endogenous inflammatory signal in the human uterus. Feedback loops purportedly limit the inflammatory decidual response to 2–4 days (*Salker et al., 2012*). The next decidual phase coincides with embedding of the implanted embryo into the stroma. At this stage, fully differentiated decidual cells, which are now tightly adherent and possess gap junctions (*Laws et al., 2008*), form an immune privileged matrix around the semi-allogenic conceptus (*Erlebacher, 2013*). In the absence of implantation, falling progesterone levels trigger a second inflammatory decidual response which, upon recruitment and activation of leukocytes, leads to tissue breakdown, focal bleeding and menstrual shedding of the superficial endometrial layer. Scar-free tissue repair involves activation of mesenchymal stem-like cells (MSCs) and epithelial progenitor cells that reside in the basal layer (*Evans et al., 2016*). Following menstruation, rising follicular E2 levels drive rapid tissue growth, which over ~10 days increases the thickness of the endometrium several-fold. Clinically, suboptimal endometrial growth is strongly associated with reproductive failure (*Yuan et al., 2016*); but how MSC activation followed by intense proliferation is linked to the decidual process is unclear.

FOXO1 is a core decidual transcription factor that controls cell cycle exit of EnSCs in response to differentiation signals and activates expression of decidual marker genes, such as *PRL* and *IGFBP1* (*Park et al., 2016*; *Takano et al., 2007*). Here we demonstrate that FOXO1 also induces acute senescence in a subpopulation of EnSCs. We show that the senescence-associated secretory phenotype (SASP) drives the initial auto-inflammatory decidual response linked to endometrial receptivity and provide evidence that uNK cells target and eliminate senescent decidual cells as the cycle progresses. Our findings reveal a hitherto unrecognized role for acute cellular senescence in endometrial remodeling at the time of embryo implantation; and suggest a major role for uNK cells in maintaining tissue homeostasis from cycle to cycle.

## Results

### Decidualization induces acute senescence in a subpopulation of EnSCs

To determine if cycling human endometrium harbor dynamic populations of senescent cells, we first stained primary EnSC cultures for senescence-associated β-galactosidase (SAβG) activity. At passage 1 (P1), SAβG$^+$ cells were detectable in variable numbers in different cultures (*Figure 1A*). Strikingly, the number of SAβG$^+$ cells increased markedly upon decidualization with 8-bromo-cAMP and medroxyprogesterone acetate (MPA, a progestin). Typically, SAβG$^+$ cells formed islets surrounded by SAβG$^-$ EnSCs in differentiating cultures (*Figure 1A*). Quantitative analysis confirmed a time-dependent increase in SAβG activity upon decidualization (*Figure 1B*). The abundance of SAβG$^+$ cells in undifferentiated cultures declined upon passaging of cells (*Figure 1—figure supplement 1A*). Initially, this was paralleled by a reduction in SAβG activity, which was reversed at later passages (P6) (*Figure 1—figure supplement 1B*), presumably reflecting emerging replicative exhaustion of EnSCs (*Figure 1—figure supplement 1C*). However, even after ~60 days in continuous culture, exposure of EnSCs to a deciduogenic stimulus enhanced SAβG activity and triggered the appearance of SAβG$^+$ cells (*Figure 1—figure supplement 1A and B*).

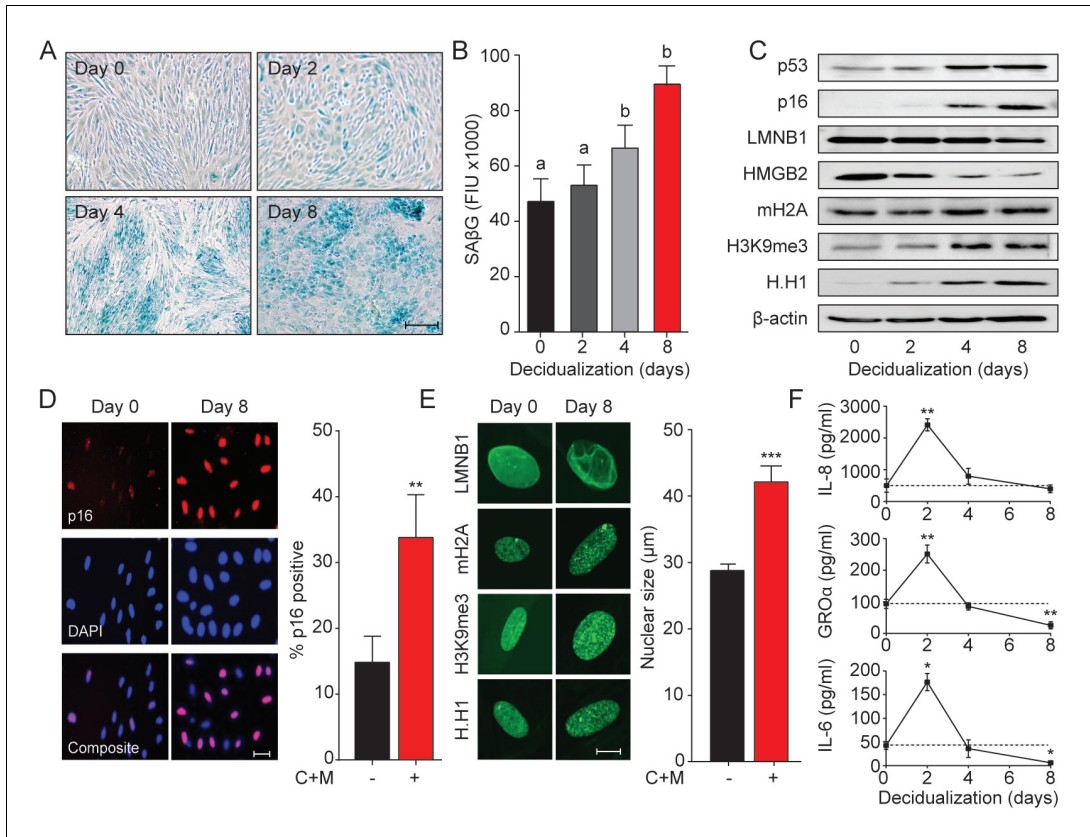

**Figure 1.** Decidualization induces acute senescence in a subpopulation of EnSCs. (A) Representative SAβG staining in undifferentiated EnSCs (Day 0) or cells decidualized for the indicated time points with 8-bromo-cAMP and MPA. Scale bar = 100 μm. (B) SAβG activity, expressed in fluorescence intensity units (FIU), in undifferentiated EnSCs (day 0) or cells decidualized for the indicated time points. (C) Representative Western blot analysis of p53, p16, LMNB1, HMGB2, mH2A, H3K9me3 and H.H1 levels in undifferentiated EnSCs and cells decidualized for the indicated time points. β-actin served as a loading control. (D) Left panel: representative immunofluorescence staining for p16 expression in undifferentiated cells and cells decidualized for 8 days. Nuclei were counterstained with DAPI. Scale bar = 50 μm. Right panel: percentage of p16+ cells. (E) Left panel: representative confocal microscopy images of undifferentiated (Day 0) or decidualized (Day 8) EnSCs immune-probed for LMNB1, mH2A, H3K9me3 and H.H1. Scale bar = 10 μm. Right panel: nuclear size of undifferentiated EnSCs (n = 48) and of cells first decidualized for 8 days with 8-br-cAMP and MPA (C + M) (n = 48) was measured in three primary cultures. (F) Secretion of IL-8, GROα, and IL-6 was measured in the supernatant of primary EnSCs collected every 48 hr over an 8 day decidualization time-course. Data are mean ±SEM of 3 biological replicates unless stated otherwise. **p<0.01, ***p<0.001. Different letters above the error bars indicate that those groups are significantly different from each other at p<0.05.

DOI: https://doi.org/10.7554/eLife.31274.002

The following source data and figure supplement are available for figure 1:

**Source data 1.** Decidualization induces acute senescence in a subpopulation of EnSCs.
DOI: https://doi.org/10.7554/eLife.31274.004

**Figure supplement 1.** Decidualization-associated acute senescence in primary EnSCs.
DOI: https://doi.org/10.7554/eLife.31274.003

Although SAβG activity is a commonly used biomarker for senescent cells, it lacks specificity (*Matjusaitis et al., 2016*). Hence, we examined the expression of other putative senescence markers in undifferentiated and decidualizing EnSCs. Many senescence signals converge onto the tumor suppressor protein p53 (p53) and induce the expression of cyclin-dependent kinase (CDK) inhibitors, leading to proliferative arrest and cell cycle exit (*Muñoz-Espín and Serrano, 2014*; *van Deursen, 2014*). We reported previously that downregulation of MDM2 proto-oncogene, an E3 ubiquitin ligase, stabilizes p53 in differentiating EnSCs (*Pohnke et al., 2004*). Western blot analysis showed

that p53 stabilization upon decidualization is paralleled by upregulation of p16[Ink4a] (p16; *Figure 1C*), a CDK inhibitor presumed specific for senescence. Notably, confocal microscopy revealed that induction of p16 upon decidualization is confined to a subpopulation of EnSCs (*Figure 1D*). Loss of lamin B1 (LMNB1) and high mobility group box 2 (HMGB2) drives many of the chromatin and epigenetic changes that underpin cellular senescence (*Aird et al., 2016*; *Sadaie et al., 2013*). Decidualization resulted in downregulation of both effector proteins and a reciprocal increase in the histone H2A variant macroH2A (mH2A) and trimethylated lysine 9 on histone H3 (H3K9Me3) (*Figure 1C* and *Figure 1—figure supplement 1D*). This histone variant and modification are involved in senescence-associated heterochromatin formation (SAHF). Unexpectedly, the nucleosome linker histone H1 (H. H1), which purportedly is lost in senescent cells (*Funayama et al., 2006*), was upregulated upon decidualization (*Figure 1C* and *Figure 1—figure supplement 1D*). These observations were confirmed by immunofluorescence confocal microscopy (*Figure 1E*, left panel), which also revealed that the nuclei of EnSC increase in size (~40%) upon decidualization (*Figure 1E*, right panel). Next, we examined if the transient inflammatory decidual response also encompasses secreted factors typical of the canonical senescence associated secretory phenotype (SASP). As shown in *Figure 1F*, secretion of IL-8 (CXCL8), GROα (CXCL1), and IL-6 peaked transiently on day 2 of decidualization and returned to baseline by day 4. By day 8, the level of secretion of GROα and IL-6 was lower than that observed in undifferentiated cultures.

Taken together, the data reveal striking similarities between cellular senescence and differentiation of EnSCs into decidual cells. However, only a subpopulation of EnSCs became strongly SAβG[+] or expressed p16 upon decidualization. Further, while SASP is often a sustained response, decidual inflammation is temporally restricted to 2–4 days.

## Temporal regulation of senescent cell populations in cycling endometrium

To extrapolate these observations to the in vivo situation, protein lysates from whole endometrial biopsies were subjected to Western blot analysis. As the cycle progresses from the proliferative to the secretory phase, the abundance of p53, p16, LMNB1, HMBG2, mH2A and H3K9me3 in the endometrium mimicked the changes observed in decidualizing EnSC cultures (*Figure 2A*). Further, analysis of snap-frozen biopsies showed a sharp increase in SAβG activity upon transition from proliferative to early-secretory endometrium with levels peaking in the late-luteal phase (*Figure 2B*). Disintegration of the stromal compartment upon cryosectioning of frozen tissue samples precluded a meaningful analysis of SAβG[+] cells. Instead, we used immunohistochemistry to examine the abundance and tissue distribution of p16[+] cells in 308 formalin-fixed endometrial biopsies obtained during the peri-implantation window, i.e. 6 to 12 days after the luteinizing hormone (LH) surge (*Figure 2C*). The statistical distribution of p16[+] cells in the glandular epithelium, luminal epithelium and stromal compartment is presented as a centile graph (*Figure 2D*). Interestingly, p16[+] cells were most prevalent in both the glandular and luminal epithelium at LH + 10 and +11, which coincides with the onset of the late-luteal phase of the cycle. The relative abundance of p16[+] cells was ~10 fold higher in the luminal compared to the glandular compartment. Typically, stretches of p16[+] cells were interspersed by p16[-] cells in the luminal epithelium (*Figure 2C*). By contrast, p16[+] cells gradually increased in the stromal compartment during the mid-luteal phase and this was accelerated in late-luteal endometrium. Occasionally, swirls of p16[+] cells were observed in the stroma, seemingly connecting the deeper regions of the endometrium to p16[+] luminal epithelial cells (*Figure 2C*). Collectively, the data indicate that the endometrium harbors dynamic and probably spatially organized populations of senescent cells during the luteal phase of the cycle.

## FOXO1 drives EnSC differentiation and senescence

To gain insight into the mechanism that drives decidual senescence, SAβG activity was measured in cultured EnSCs treated for 8 days with either 8-bromo-cAMP, MPA or a combination. Both 8-bromo-cAMP and MPA were required for significant induction of SAβG activity ($p < 0.05$) (*Figure 3A*). In differentiating EnSCs, cAMP and progesterone signaling converge on FOXO1, a core decidual transcription factor responsible for cell cycle arrest and induction of decidual marker genes, such as *PRL* and *IGFBP1* (*Takano et al., 2007*). Interestingly, FOXO1 was also shown to induce cellular senescence of ovarian cancer cells treated with progesterone (*Diep et al., 2013*). siRNA-

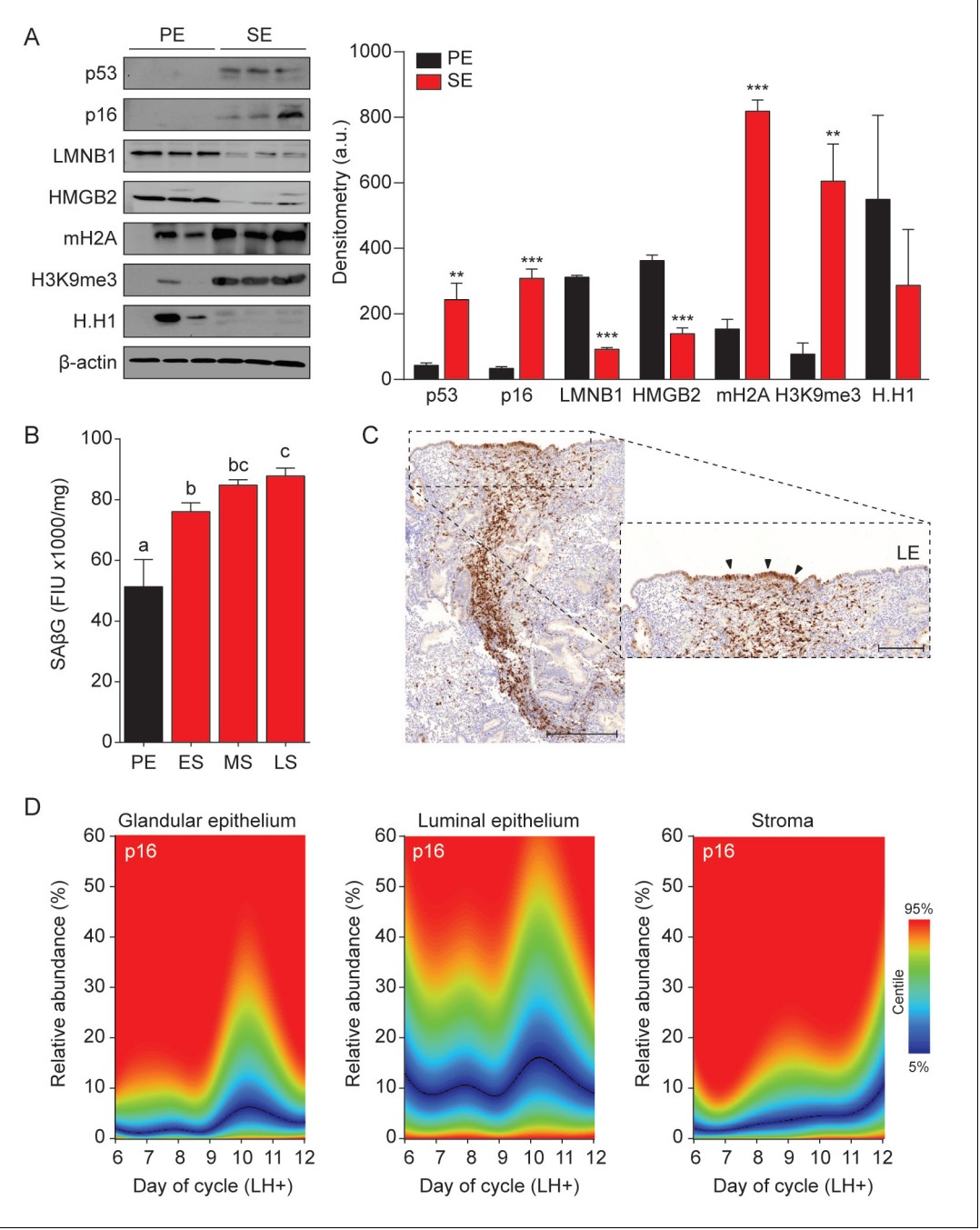

**Figure 2.** Senescent cells in cycling human endometrium. (**A**) Left panel: representative Western blot analysis of p53, p16, LMNB1, HMGB2, mH2A, H3K9me3 and H.H1 levels in whole tissue biopsies from proliferative endometrium (PE) and secretory endometrium (SE). β-actin served as a loading control. Right panel: protein levels quantified relative to β-actin by densitometry and expressed as arbitrary units (a.u.). (**B**) SAβG activity, expressed in fluorescence intensity units (FIU)/mg protein, was measured in biopsies from proliferative endometrium (PE; n = 7), early-secretory (ES; n = 9), mid-secretory (MS; n = 38) and late-secretory (LS; n = 19) endometrium. (**C**) Immunohistochemistry demonstrating distribution of p16+ cells in the stromal compartment and luminal epithelium. Scale bars = 200 μm. (**D**) The abundance of p16+ cells during the luteal phase in glandular epithelium, luminal epithelium and stroma compartment was analyzed by color deconvolution using ImageJ software in 308 LH-timed endometrial biopsies (average 48 samples per time point; range: 22 to 69). The centile graphs depict the distribution of p16+ cells across the peri-implantation window in each cellular compartment. Color key is on the right. Data are mean ±SEM of 3 biological replicates unless stated otherwise. **p<0.01, ***p<0.001. Different letters above the error bars indicate that those groups are significantly different from each other at p<0.05.

*Figure 2 continued on next page*

*Figure 2 continued*

DOI: https://doi.org/10.7554/eLife.31274.005

The following source data is available for figure 2:

**Source data 1.** Senescent cells in cycling human endometrium.
DOI: https://doi.org/10.7554/eLife.31274.006

mediated knockdown of FOXO1 in EnSCs not only abolished the induction of *PRL* and *IGFBP1* (*Figure 3—figure supplement 1A*) but also the surge in IL-8, GROα, and IL-6 secretion upon treatment of cultures with 8-bromo-cAMP and MPA (*Figure 3B*). After 8 days of decidualization, FOXO1 knockdown was less efficient but nevertheless sufficient to significantly blunt SAβG activity (p<0.05) (*Figure 3C*). Several components of the SASP have been implicated in autocrine/paracrine propagation of senescence, including IL-8 acting on CXCR2 (IL-8 receptor type B) (*Acosta et al., 2008*). In agreement, decidualization of EnSCs in the presence of SB265610, a potent CXCR2 inhibitor, attenuated SAβG activity in a dose-dependent manner (*Figure 3D*). Conversely, recombinant IL-8 upregulated SAβG activity in undifferentiated EnSCs in a concentration-dependent manner (*Figure 3D*), although spatial organization of SAβG$^+$ cells into 'islets' was not observed (*Figure 3—figure supplement 1B*). Further, siRNA-mediated *CXCL8* (coding IL-8) knockdown in undifferentiated EnSCs not only blunted the surge in IL-8 secretion upon decidualization (*Figure 3—figure supplement 1B*), but also abolished the increase in SAβG activity (*Figure 3E*). Unexpectedly, IL-8 knockdown compromised the induction of *PRL* and *IGFBP1* in cultures treated with 8-bromo-cAMP and MPA (*Figure 3F*), indicating that autocrine/paracrine signals involved in EnSC differentiation also drive decidual senescence.

In an attempt to block decidual senescence selectively, EnSCs were differentiated in the presence or absence of the mTOR inhibitor rapamycin, a pharmacological repressor of replicative senescence (*Demidenko et al., 2009*). Rapamycin prevented expansion of SAβG$^+$ cells upon decidualization but also abolished expression of *PRL* and *IGFBP1* (*Figure 3—figure supplement 1C and D*). By contrast, withdrawal of 8-bromo-cAMP and MPA from cultures first decidualized for 8 days reversed the induction of decidual marker genes (*Figure 3G*), albeit without impacting on either SAβG activity or expression of p53, p16, LMNB1 and HMGB1 (*Figure 3H*). Taken together, the data demonstrate that FOXO1 drives both differentiation and senescence of distinct EnSC subpopulations in an IL-8 dependent manner; however, while expression of differentiation markers requires continuous cAMP and progestin signaling, the senescent phenotype does not.

## Pleiotropic functions of senescent decidual cells

The abundance of SAβG$^+$ cells in undifferentiated cultures correlated closely with the number of SAβG$^+$ cells upon decidualization (Pearson's *r* = 0.97, p<0.0001) (*Figure 4—figure supplement 1A*). A congruent correlation was apparent upon measuring SAβG activity in paired undifferentiated and decidualizing cultures (Pearson's *r* = 0.91, p<0.0001) (*Figure 4A*), inferring that senescent decidual cells arise from stressed (presenescent) EnSCs. Hence, we tested if decidualization-associated senescence could be blocked by pretreating undifferentiated cultures with senolytic drugs. Exposure of primary EnSCs for 72 hr to increasing concentrations of ABT-263 (Navitoclax), a BCL-X$_L$ inhibitor (*Zhu et al., 2016*), had no discernible impact on the induction of SAβG activity upon decidualization (*Figure 4—figure supplement 1B*). By contrast, pretreatment of primary cultures with dasatinib, a broad-spectrum tyrosine kinase inhibitor (*Childs et al., 2017*; *Zhu et al., 2015*), inhibited SAβG activity upon decidualization in a dose-dependent manner (*Figure 4—figure supplement 1C*). Conversely, to increase the abundance SAβG$^+$ cells, undifferentiated EnSCs were treated with the CDK4/CDK6 inhibitor palbociclib (PD0332991), a functional p16 mimetic (*Mosteiro et al., 2016*). Dose-response analyses showed that treatment with palbociclib for 4 days was sufficient to increase SAβG activity in undifferentiated EnSCs to the level observed in cells decidualized with 8-bromo-cAMP and MPA for 8 days (*Figure 4—figure supplement 1C*). Notably, neither dasatinib nor palbociclib pretreatment impacted on the induction of *PRL* or *IGFBP1* upon decidualization (*Figure 4C*). However, dasatinib pretreatment markedly blunted the surge in IL-8, IL-6, and GROα secretion that characterizes the initial decidual phase. By contrast, this auto-inflammatory decidual response was amplified in response to palbociclib pretreatment (*Figure 4D*).

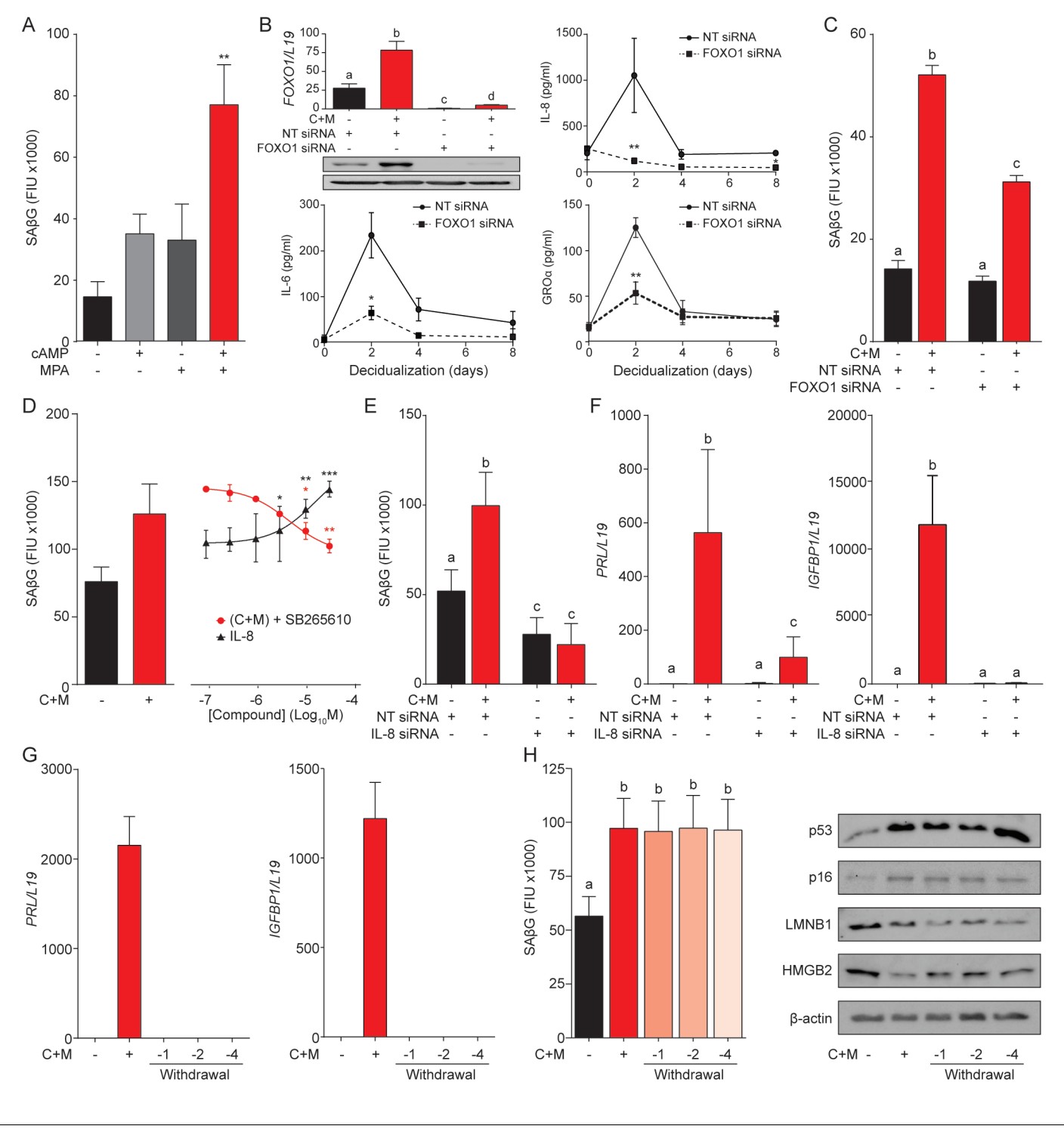

**Figure 3.** A FOXO1/IL-8 axis drives EnSC differentiation and senescence. (**A**) SAβG activity in EnSCs either undifferentiated, or decidualized for 8 days with 8-bromo-cAMP, MPA, or a combination. (**B**) Top left panel: *FOXO1* mRNA levels in undifferentiated EnSCs and cells treated with 8-br-cAMP and MPA (C + M) following transfection with non-targeting (NT) or FOXO1 siRNA. Other panels: Secretion of IL-8, IL-6 and GROα was measured following FOXO1 knockdown in the supernatant of primary EnSCs every 48 hr over an 8 day decidualization time-course. (**C**) SAβG activity in EnSCs following transfection with NT or FOXO1 siRNA. The cultures either remain untreated or decidualized for 8 days. (**D**) SAβG activity in undifferentiated EnSCs treated for 8 days with increasing concentrations of recombinant IL-8 and in cells decidualized for 8 days in the presence of increasing concentrations of the CXCR2 antagonist, SB265610. (**E**) SAβG activity in EnSCs following transfection with IL-8 siRNA. The cultures either remain untreated or decidualized for 8 days. (**F**) *PRL* and *IGFBP1* transcript levels in EnSCs following transfection with IL-8 siRNA. The cultures either remain untreated or
*Figure 3 continued on next page*

*Figure 3 continued*

decidualized for 8 days. (G) *PRL* and *IGFBP1* expression in undifferentiated EnSCs, cells decidualized for 8 days, and upon withdrawal of 8-br-cAMP and MPA (C + M) for the indicated days. (H) Left panel: SAβG activity in undifferentiated EnSCs, cells decidualized for 8 days, and following withdrawal of C + M for the indicated days. Right panel: representative Western blot analysis of p53, p16, LMNB1 and HMGB2 levels in undifferentiated EnSCs, cells decidualized for 8 days, and following withdrawal of C + M for the indicated days. β-actin served as a loading control. Data are mean ±SEM of 3 biological replicates unless stated otherwise. *p<0.05, **p<0.01 and ***p<0.005. Different letters above the error bars indicate that those groups are significantly different from each other at p<0.05.

DOI: https://doi.org/10.7554/eLife.31274.007

The following source data and figure supplement are available for figure 3:

**Source data 1.** A FOXO1/IL-8 axis drives EnSC differentiation and senescence.
DOI: https://doi.org/10.7554/eLife.31274.009
**Figure supplement 1.** EnSC differentiation and senescence is driven by FOXO1, IL-8 and mTOR.
DOI: https://doi.org/10.7554/eLife.31274.008

In other cell systems, transient - but not prolonged - exposure to SASP has been shown to promote tissue rejuvenation by reprogramming committed cells into stem-like cells (*Ritschka et al., 2017*). We reasoned that SASP-dependent tissue rejuvenation during the window of implantation may be relevant for the transition of the cycling endometrium into the decidua of pregnancy. To test this hypothesis, we examined the clonogenic capacity of paired undifferentiated cells and cells decidualized for 8 days. Analysis of 12 primary cultures demonstrated that decidualization is associated with a modest but significant increase in colony-forming cells (p<0.05), although the response varied between primary cultures (*Figure 4E*). However, pretreatment of undifferentiated cultures with dasatinib or palbociclib consistently increased or decreased the clonogenic capacity of decidualizing cultures, respectively (*Figure 4F*). Likewise, rapamycin also depleted decidualizing EnSC cultures of clonogenic cells (*Figure 4—figure supplement 1D*). Taken together, the data suggest that senescent decidual cells produce a transient inflammatory environment that not only contributes to endometrial receptivity but also increases tissue plasticity prior to pregnancy.

## Immune clearance of senescent decidual cells

Recognition and elimination of senescent cells by immune cells, especially NK cells, play a pivotal role in tissue repair and homeostasis (*Iannello and Raulet, 2013*; *Krizhanovsky et al., 2008*). During the luteal phase, uNK cells, characterized by their CD56[bright] cell surface phenotype (*Figure 5—figure supplement 1A*), are by far the dominant endometrial leukocyte population. Analysis of a large number of LH-timed endometrial biopsies (n = 1,997) demonstrated that the abundance of CD56[+] uNK cells in the endometrial stromal compartment increases on average 3-fold between LH + 5 and +12; although inter-patient variability was marked (*Figure 5A*, left panel). The heatmap in *Figure 5A* (right panel) depicts the uNK cell centiles across the peri-implantation window. Notably, uNK cells often appear to amass in edematous areas that are relatively depleted of stromal cells, especially during the transition from the mid- to late-luteal phase of the cycle (*Figure 5B*, left panel). Quantitative analysis of 20 biopsies obtained between LH + 9 and LH + 11 confirmed the inverse correlation between the density of uNK cells and endometrial stromal cells (*Figure 5B*, right panel). In co-culture, uNK cells isolated from secretory endometrium had no impact on proliferation or viability of undifferentiated EnSCs (*Figure 5—figure supplement 1B*, left panel). By contrast, co-culture of uNK cells with EnSCs first decidualized for 8 days resulted in loss of cell viability. Visually, uNK cells transformed the monolayer of decidual cells into a honeycomb pattern with cell-free islets (*Figure 5C*). Selective targeting and clearance of decidual cells by uNK cells was also captured by time-lapse microscopy (see *Video 1*).

These observations suggested that uNK cells actively eliminate senescent EnSCs but only upon decidualization. In agreement, co-culturing of uNK cells with EnSCs eliminated the induction of SAβG activity upon decidualization without affecting basal activity in undifferentiated cells (*Figure 5D*). Two independent mechanisms underpin NK cell-mediated clearance of stressed cells (*Sagiv et al., 2013*). First, binding of the NK cell surface ligands TRAIL and FAS ligand (FasL) to the corresponding receptors on target cells can lead to caspase activation and cell death. However, incubation of primary EnSCs with increasing concentrations of FasL or TRAIL had no impact on SAβG activity in either undifferentiated or decidualizing cells (*Figure 5—figure supplement 1C*),

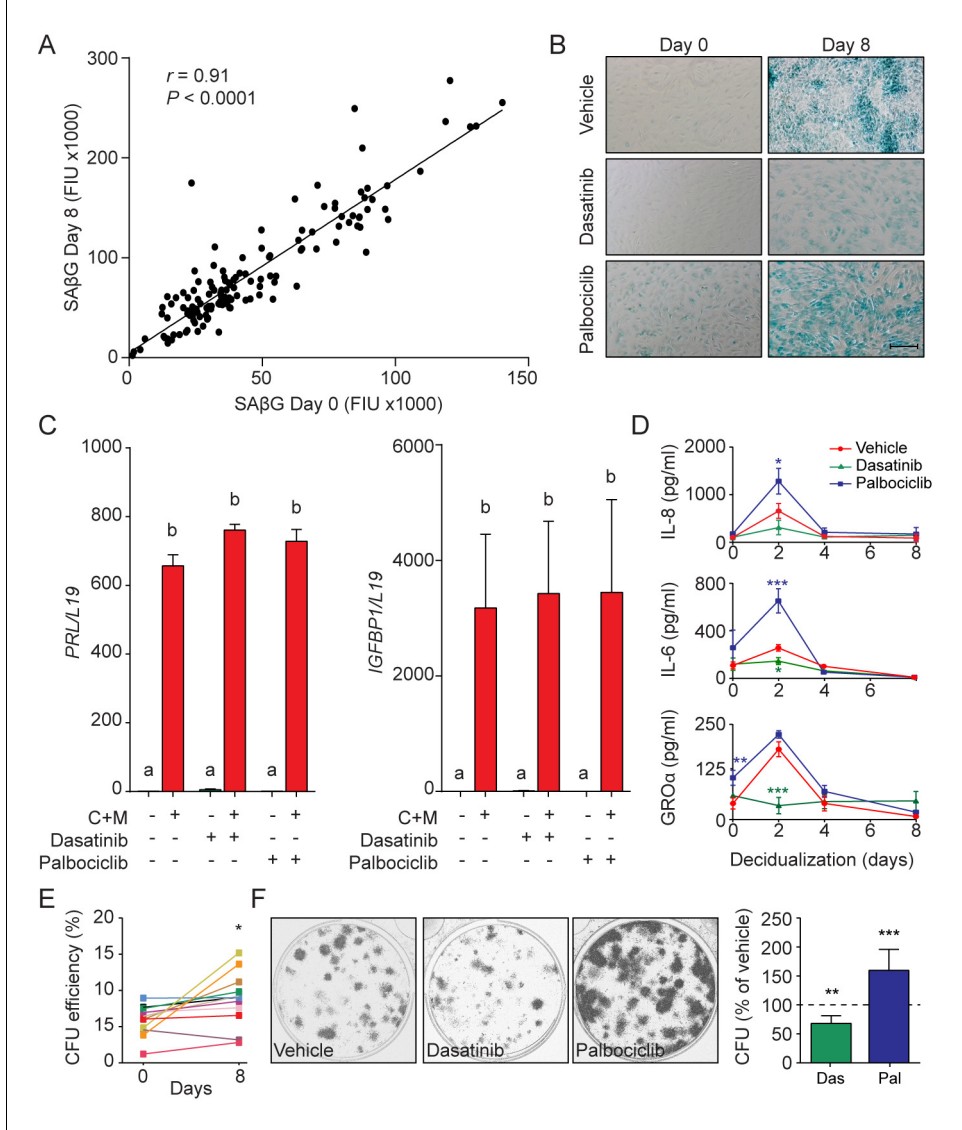

**Figure 4.** Functions of senescent decidual cells. (**A**) Pearson's correlation analysis of SAβG activity in 75 matched undifferentiated primary cultures and cultures decidualized for 8 days. (**B**) Representative SAβG staining in undifferentiated (Day 0) and decidualizing EnSCs (Day 8) following 4 days of pretreatment with vehicle, dasatinib (250 nM) or palbociclib (1 μM). Scale bar = 100 μm. (**C**) *PRL* and *IGFBP1* mRNA expression in response to pretreatment with vehicle, dasatinib or palbociclib. The cultures then remained undifferentiated or were decidualized for 8 days. (**D**) IL-8, IL-6 and GROα secretion was measured every 48 hr in the supernatant of primary EnSCs decidualized for the indicated time-points following pretreatment with vehicle, dasatinib or palbociclib. (**E**) Colony forming unit (CFU) activity in paired EnSC cultures that either remain undifferentiated (Day 0) or were decidualized for 8 days (n = 10). (**F**) Left panel: representative clonogenic assays established from EnSC cultures first pretreated with vehicle, dasatinib or palbociclib and then decidualized for 8 days. Right panel: CFU activity in EnSC cultures first pretreated with vehicle, dasatinib or palbociclib and then decidualized for 8 days. Data are mean ±SEM of 3 biological replicates unless stated otherwise. *p<0.05, **p<0.01 and ***p<0.001. Different letters above the error bars indicate that those groups are significantly different from each other at p<0.05.

DOI: https://doi.org/10.7554/eLife.31274.010

The following source data and figure supplement are available for figure 4:

**Source data 1.** Functions of senescent decidual cells.
DOI: https://doi.org/10.7554/eLife.31274.012

**Figure supplement 1.** Modulation of senescence in EnSC cultures.
DOI: https://doi.org/10.7554/eLife.31274.011

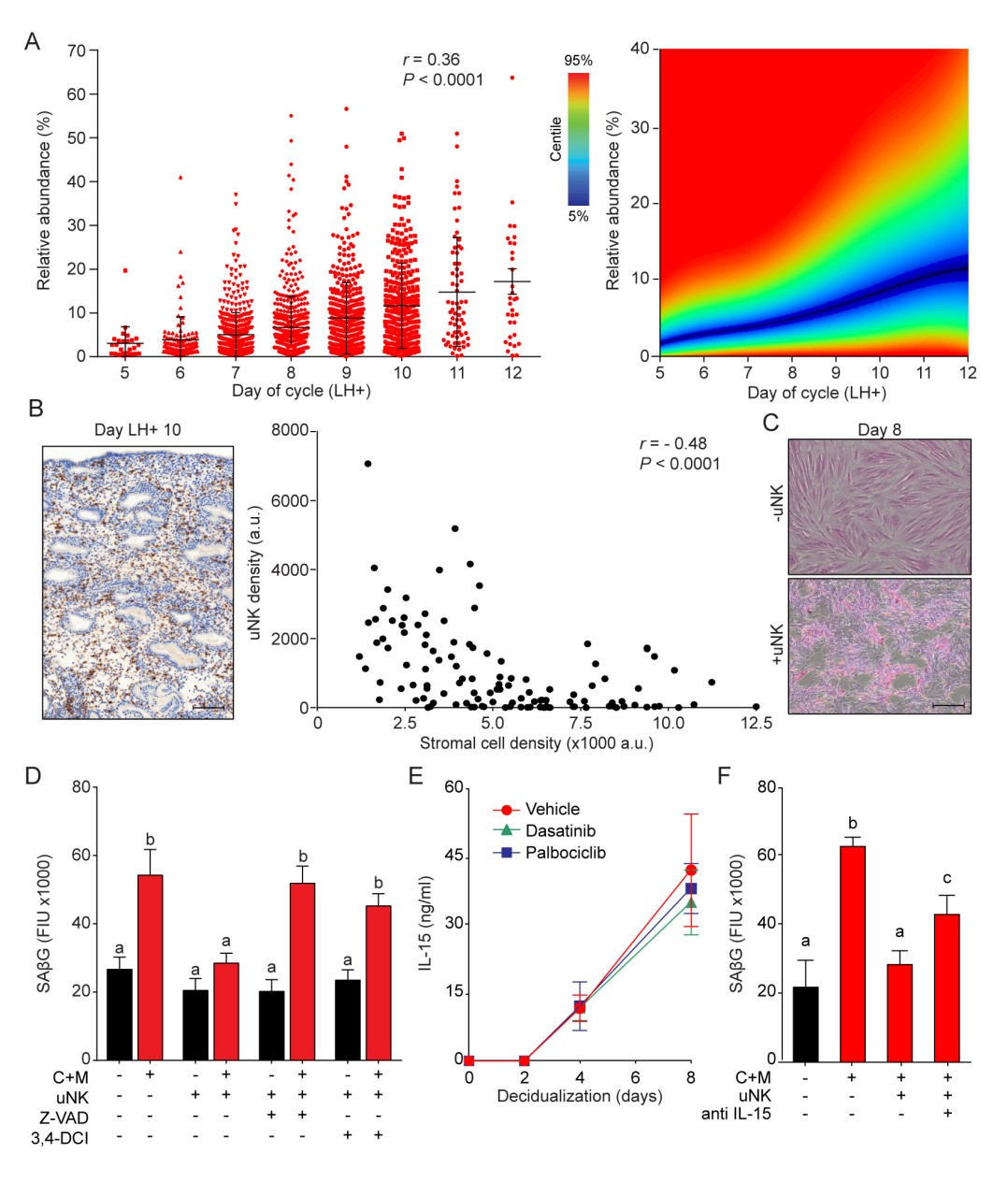

**Figure 5.** uNK cell mediated immune surveillance and clearance of senescent cells. (**A**) Left panel: uNK cell density in the subluminal stroma was quantified using a standardized immunohistochemistry protocol in LH timed endometrial biopsies (n = 1,997). Right panel: corresponding centile graph. Color code on the left. (**B**) Left panel: example of the tissue distribution of CD56$^+$ uNK cells (brown staining) at LH + 10. Scale bar = 250 µm. Right panel: Pearson's correlation analysis of stromal cell and uNK cell densities. A total of 80 randomly selected images from 20 biopsies were analyzed. (**C**) Representative images of an eosin stained primary culture decidualized for 8 days incubated for 18 hr with or without uNK cells isolated from luteal phase endometrium. Scale bar = 100 µm. (**D**) SAβG activity in undifferentiated or day eight decidualized EnSCs co-cultured with or without uNK cells in the presence or absence of the apoptosis inhibitor Z-VAD-FMK (Z-VAD, 10 µM) or the granzyme activity inhibitor 3,4-DCI (25 µM). (**E**) Secretion of IL-15 secretion was measured every 48 hr in the supernatant of primary EnSCs decidualized for the indicated time-points following pretreatment with vehicle, dasatinib (250 nM) or palbociclib (1 µM). (**F**) SAβG activity in undifferentiated or day eight decidualized EnSCs co-cultured with or without uNK cells in the presence or absence of an IL-15 blocking antibody (1 µg/ml). Data are mean ±SEM of 3 biological replicates unless stated otherwise. Different letters above the error bars indicate that those groups are significantly different from each other at p<0.05.

*Figure 5 continued on next page*

*Figure 5 continued*

DOI: https://doi.org/10.7554/eLife.31274.013

The following source data and figure supplement are available for figure 5:

**Source data 1.** uNK cell mediated immune surveillance and clearance of senescent cells.
DOI: https://doi.org/10.7554/eLife.31274.015
**Figure supplement 1.** uNK cell mediated immune surveillance and clearance of senescent cells.
DOI: https://doi.org/10.7554/eLife.31274.014

inferring that death receptor activation is not required for uNK cell-mediated senolysis. The second mechanism involves secretion by activated NK cells of granules containing perforin and granzyme (A, B). Perforin forms pores in the plasma membrane of target cells and triggers apoptosis upon release of granzyme into the cytoplasm (*Chowdhury and Lieberman, 2008*). As shown in *Figure 5D* and *Figure 5—figure supplement 1B*, both the pan-caspase inhibitor Z-VAD-FMK and the granzyme B inhibitor 3,4-Dichloroisocoumarin (3,4-DCI) negated the impact of uNK cells on SAβG activity and cell viability in decidualizing cultures. To explore why uNK cell-mediated clearance of SAβG$^+$ EnSCs is restricted to decidualizing cultures, we focused on IL-15, a pivotal cytokine that regulates NK cell proliferation and activation (*Marçais et al., 2014*). IL-15 secretion was below the level of detection in undifferentiated cells but, after a lag-period of 2 days, rose markedly upon decidualization of EnSCs in a time-dependent manner (*Figure 5E*). Notably, pretreatment of cultures with dasatinib or palbociclib had no impact on impact on IL-15 secretion, suggesting that differentiated EnSCs orchestrate the uNK-mediated clearance of their senescent counterparts (*Figure 5E*). Incubation of co-cultures with an IL-15 blocking antibody antagonized, at least partly, uNK cell-mediated clearance of senescent decidual cells (*Figure 5F*). Furthermore, a blocking antibody against NKG2D (Natural Killer Group 2D), a receptor expressed by all human NK cells and CD8-positive T cells that binds surface ligands expressed on stressed cells (*Sagiv et al., 2016*), also attenuated uNK cell-mediated killing of senescent decidual cells (*Figure 5—figure supplement 1D*).

## Tissue homeostasis

Our findings indicate that endometrial homeostasis during the luteal phase is dependent on balancing induction and clearance of senescent decidual cells. We speculated that this process is *a priori* dynamic, which should be reflected in varying numbers of uNK cells in different cycles. As proof of concept, we quantified uNK cells in biopsies from three patients obtained around the same time in the mid-luteal phase (±1 day) in three different cycles. As shown in *Figure 6A*, the abundance of uNK cells in the subluminal endometrial stroma can vary profoundly between cycles. As levels both rose and fell, the observed inter-cycle changes in uNK cell density are unlikely triggered by the tissue injury caused by the biopsy, although an impact on the magnitude of change cannot be excluded. Additional examples of uNK cell fluctuations in two consecutive cycles are presented in *Figure 6—figure supplement 1*.

## Discussion

In contrast to chronic senescence associated with organismal ageing, acute senescence is a tightly orchestrated biological process implicated in embryo development, wound healing and tissue repair. Typically, acute senescent cells produce a context-specific SASP with defined paracrine functions and self-organize their elimination by various immune cells (*van Deursen, 2014*). Here we provide evidence that acute decidual senescence is a pivotal process that coordinates acquisition of a receptive phenotype

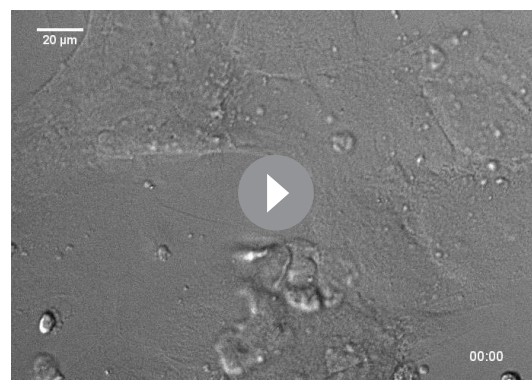

**Video 1.** EnSCs and uNK co-cultures. Time-lapse microscopy of 8 day decidualized EnSCs and uNK co-cultures. Images captured at a rate of 1 frame per 10 min. Time (hours) and scale as indicated.
DOI: https://doi.org/10.7554/eLife.31274.016

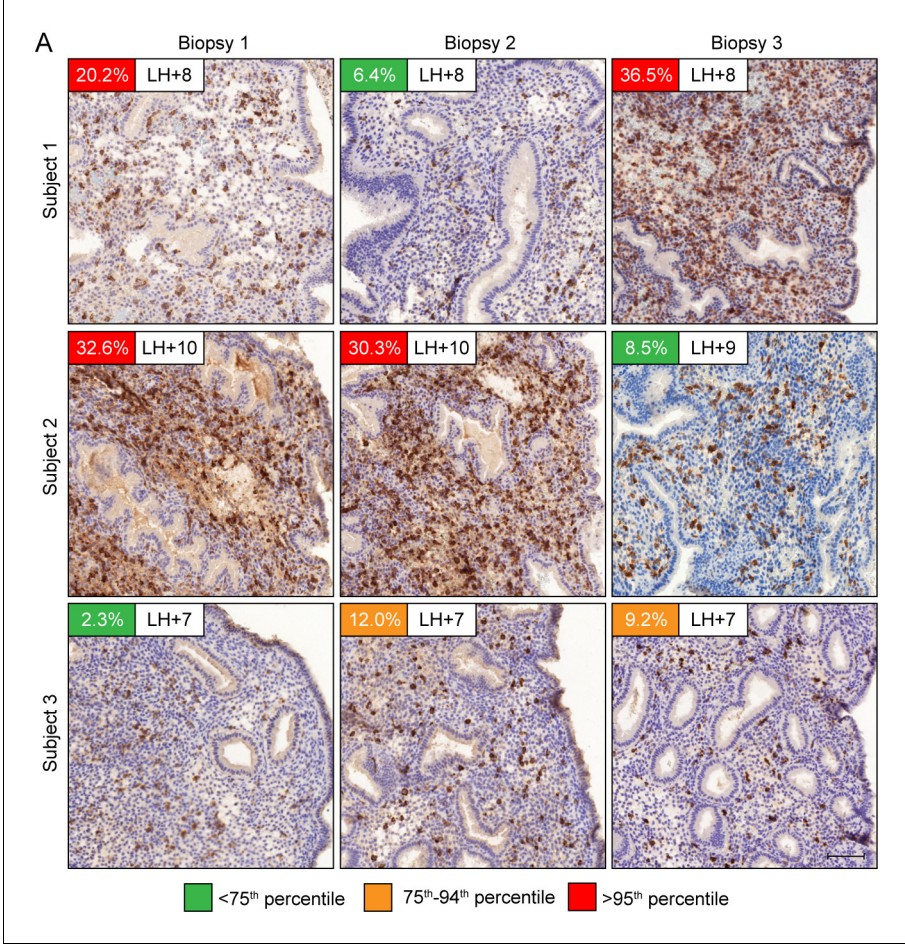

**Figure 6.** Dynamic inter-cycle fluctuations in uNK cell levels. CD56 immunohistochemistry of LH-timed endometrial biopsies obtained in three different cycles in three subjects. The day of the biopsy and the percentage of CD56$^+$ uNK cells versus stromal cells are indicated. The color of the box indicates the percentile range of uNK when adjusted for the day of biopsy. Scale = 200 μm.

DOI: https://doi.org/10.7554/eLife.31274.017

The following figure supplement is available for figure 6:

**Figure supplement 1.** uNK cells in repeat biopsies.

DOI: https://doi.org/10.7554/eLife.31274.018

with endometrial remodeling and rejuvenation during the implantation process. We reported previously that decidual transformation of primary EnSCs is a stepwise process that starts with a NOX4-dependent burst of free radicals and release of multiple inflammatory mediators (*Al-Sabbagh et al., 2011*; *Lucas et al., 2016b*; *Salker et al., 2012*). Exposure of the mouse uterus to this inflammatory secretome induced expression of multiple implantation genes and enabled efficient implantation of in vitro cultured mouse embryos (*Salker et al., 2012*). We now demonstrate that this nidatory decidual signal is driven foremost by acute senescence of a subpopulation of EnSCs. The close correlation between SAβG activity before and after decidualization suggests that polarization of EnSCs upon cell cycle exit into differentiating and senescent cells is not stochastic but determined by the level of replicative stress incurred by individual EnSCs during the preceding proliferative phase (*Figure 7*). Acute senescence rejuvenates the receptive endometrium through two distinct mechanisms. First, we demonstrated that blocking of decidual senescence with rapamycin or elimination of senescent cells with dasatinib result in loss of clonal MSCs in differentiating EnSC cultures. By contrast, amplification of senescence in response palbociclib treatment consistently increased the clonal cell population. These observations indicate that the decidual SASP not only 'locks in' endometrial MSCs upon

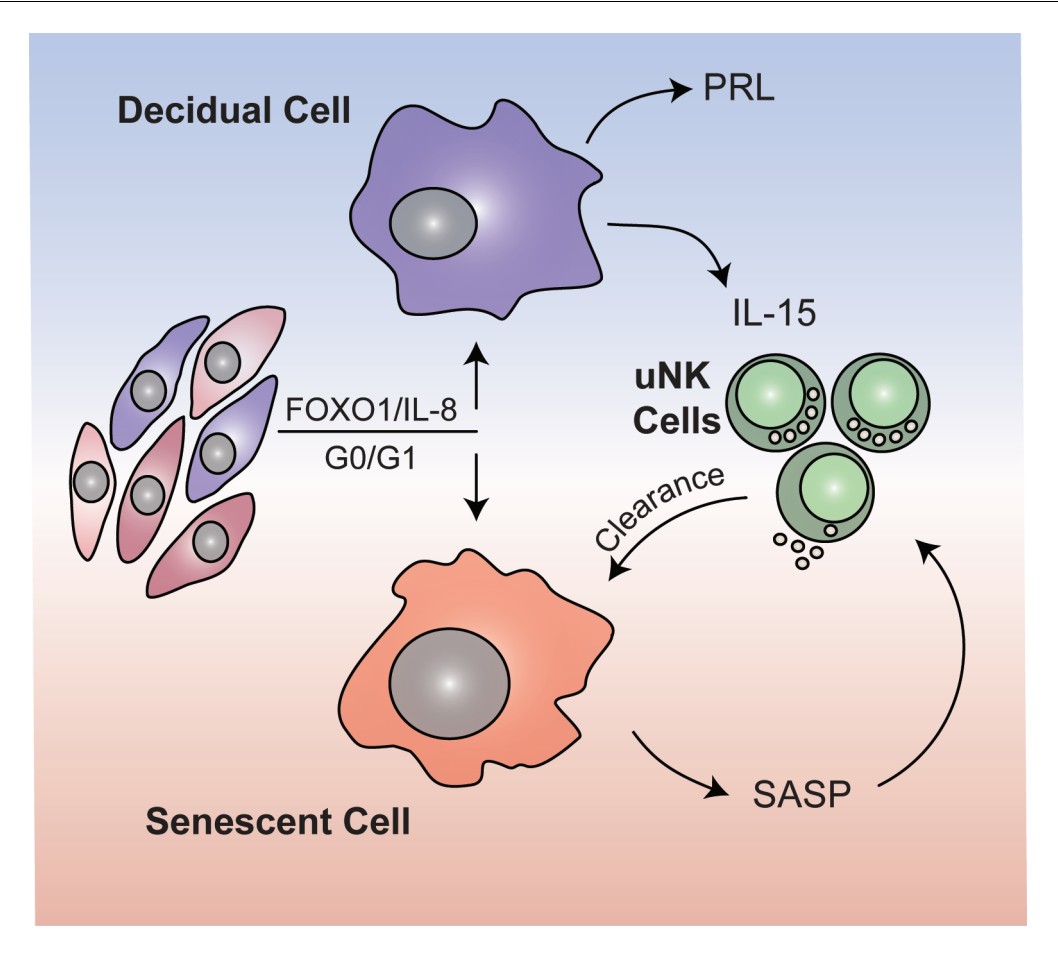

**Figure 7.** Schematic summary. We propose that rapid endometrial growth during the proliferative phase is important for implantation as it imparts replication stress in a subpopulation of EnSCs. Upon cell cycle exit at G0/G1, this subpopulation of stressed EnSCs do not differentiate into specialist decidual cells but undergo acute cellular senescence and secrete a host of inflammatory mediators (senescence associated secretory phenotype; SASP) involved in endometrial receptivity. In parallel, Il-15 secreted by differentiated decidual cells activates uNK cells, which then target and eliminate senescent cells through granule exocytosis. Systematic clearance of acutely senescent decidual cells by uNK cells not only remodels but also rejuvenates the endometrium at the time of embryo implantation.

DOI: https://doi.org/10.7554/eLife.31274.019

decidualization but, dependent on the amplitude of the inflammatory response, also de-differentiates more committed cells into clonal MSCs (*de Keizer, 2017*; *Ritschka et al., 2017*). Arguably, an adequate MSC population may be required for decidual expansion in pregnancy. Second, clearance of senescent decidual cells upon uNK cell activation ensures that the embryo embeds in a preponderance of mature decidual cells. In co-culture, uNK cell mediated clearance of SAβG⁺ cells transformed the decidual cell monolayer into a honeycomb pattern. If recapitulated in vivo, this observation suggests a role for uNK cells in creating ingresses in the tightly adherent decidual cell matrix to facilitate trophoblast invasion and anchoring of the conceptus. Compared to undifferentiated EnSCs, decidual cells are highly resistant to various stress signals, convert inactive cortisone into cortisol through the induction of 11β-hydroxysteroid dehydrogenase type 1, and protect the embryo-maternal interface from influx of T-cells by silencing genes coding for key chemokines (*Erlebacher, 2013*; *Gellersen and Brosens, 2014*). Taken together, these observations suggest that senescent decidual cells trigger a dynamic tissue reaction that ultimately results in enclosure of the conceptus into an immune-privileged decidual matrix. In pregnancy, uNK cells express senescence

markers and are proangiogenic rather than cytotoxic (*Rajagopalan and Long, 2012*). Whether prior exposure to senescent decidual cells contributes to this gestational phenotype of uNK cells is an intriguing but as yet untested possibility.

Notably, p16[+] epithelial cells were present throughout the peri-implantation window, although relatively much more so in the luminal compared to glandular epithelium. It is conceivable that p16[+] luminal epithelial cells play a role in directing the embryo to preferential sites of implantation. However, the abundance of p16[+] cells in both epithelial compartments peaked on the transition of mid- to late-luteal phase, which in turn points towards a potential role for cellular senescence in rendering the endometrium refractory to further implantation. The mechanism driving senescence of endometrial epithelial cells may differ from that in stromal cells. For example, unlike their stromal counterparts, endometrial epithelial cells express human telomerase reverse transcriptase (hTERT) and exhibit dynamic telomerase activity (*Hapangama et al., 2017*). Interestingly, hTERT expression and telomerase activity fall dramatically during the mid-luteal phase in the cycle, i.e. preceding the rise in P16[+] epithelial cells observed in this study. Telomere lengths are also shortest during the mid-secretory phase, which points to an alternative mechanism of triggering cell cycle exit and senescence of epithelial cells.

Clinically, recurrent pregnancy loss (RPL) is a distressing disorder that often remains unexplained despite extensive investigations (*Lucas et al., 2016a*). Embryonic chromosome instability accounts for a majority of sporadic failures. However, the likelihood of an underlying endometrial defect compromising the development of a euploid embryo increases with each additional failure. Nevertheless, the cumulative live birth rate following multiple miscarriages or IVF failures is high (*Lucas et al., 2016a*; *Smith et al., 2015*), which suggests that embryo-endometrial interactions are intrinsically dynamic. Our findings point towards a new paradigm that accounts for the observation that RPL does not preclude a successful pregnancy. If the level of replicative stress during the proliferative phase is efficiently counterbalanced by uNK cell-mediated clearance of senescence decidual cells during the luteal phase, implantation competence of the endometrium is assured and, in the absence of other pathology, reproductive fitness should be maximal. If not, the frequency of aberrant cycles, and thus the likelihood of reproductive failure, is predicted to increase in line with the degree of endometrial dyshomeostasis. For example, the endometrium in RPL patients is characterized by MSC deficiency, heightened cellular senescence and a prolonged and disordered decidual inflammatory response (*Lucas et al., 2016b*; *Salker et al., 2012*). Our model predicts that excessive decidual senescence can be counterbalanced by increased uNK cell proliferation and activation, thus tending towards homeostasis and leading to intermittent normal cycles. Importantly, the degree of endometrial MSC deficiency correlates with the number of previous miscarriages and, by extension, the likelihood of further failure (*Lucas et al., 2016b*). This observation provides credence to our assertion that the chance of a successful pregnancy correlates inversely with the severity of endometrial dyshomeostasis. The corollary of an intrinsic ability to balance induction and clearance of senescent cells from cycle to cycle is that the human endometrium seems refractory to ageing and maintains its function throughout the reproductive years (*Hapangama et al., 2017*).

In summary, acute senescence of a subpopulation of stromal cells upon decidualization triggers a multi-step process that transforms the cycling endometrium into a gestational tissue. Endometrial remodeling at the time of embryo implantation is controlled spatiotemporally by the level of decidual senescence and the efficacy of immune clearance.

## Materials and methods

### Key resources table

| Reagent type | Designation | Source | Identifiers |
| --- | --- | --- | --- |
| antibody | p53 | Dako | RRID:AB_2206626 |
| antibody | P16[INK4a]; p16 | Abcam | RRID:AB_10858268 |
| antibody | Lamin B1; LMN1 | Abcam | RRID:AB_443298 |
| antibody | HMBG2 | Abcam | RRID:AB_1140885 |
| antibody | macro.H2A; mH2A | Abcam | RRID:AB_2716576 |

*Continued on next page*

*Continued*

| Reagent type | Designation | Source | Identifiers |
| --- | --- | --- | --- |
| antibody | Histone H1; H.H1 | Abcam | RRID:AB_11000797 |
| antibody | H3K9me3 | Abcam | RRID:AB_306848 |
| antibody | β-actin | Sigma-Aldrich | RRID:AB_476744 |
| antibody | IL-15 | R and D Systems | RRID:AB_2124578 |
| antibody | NKG2D | R and D Systems | RRID:AB_2133263 |
| antibody | PE-conjugated anti-CD56 | Biolegend | RRID:AB_2563924 |

## Patient recruitment and sample collection

Endometrial biopsies were obtained from patients attending the Implantation Clinic, a dedicated research clinic at University Hospitals Coventry and Warwickshire (UHCW) NHS Trust, Coventry, UK. All research was undertaken with NHS National Research Ethics Committee approval (1997/5065). All biopsies were retrieved from the Arden Tissue Bank at UHCW. All participants provided written informed consent in accordance with the guidelines of the Declaration of Helsinki, 2000. Subjects were not on hormonal treatment for at least 3 months prior to the procedure. Biopsies were timed between 5–13 days post-LH surge according to home ovulation tests. Endometrial biopsies were obtained using a Wallach Endocell endometrial sampler. The device was introduced into the uterus until resistance from the fundus was felt. The piston was withdrawn to generate negative pressure and the device rotated through 360° as it was gradually withdrawn from the fundus to the cervical ostium. Biopsies were immediately portioned for use in multiple applications: snap frozen in $LN_2$, fixed in 10% formalin for immunohistochemistry and placed in 10% DMEM/F12 (see below) for cell dissociation and culture. A total of 2111 endometrial biopsies were used in this study. Patient demographics are summarized in *Supplementary file 1*.

## Cell dissociation and primary culture

Primary EnSC cultures were established as described previously in detail (*Barros et al., 2016*). Biopsy tissue was rinsed in media and finely minced. Minced tissues were incubated with collagenase type Ia (134 U/ml) (Sigma-Aldrich, Gillingham, UK) and deoxyribonuclease type 1 (156 U/ml) (Roche, Burgess Hill, UK) in 10 ml additive-free DMEM/F12 for 1 hr at 37°C with manual agitation at 20 min intervals. Digested tissue was filtered through 40 µm cell strainers to eliminate glandular clumps and flushed with 10% DMEM/F12 (see below). Cells were collected by centrifugation ($280 \times g$, 5 min, RT) and re-suspended in 10% DMEM/F12 media for culture. Media was changed 6–18 hr post-seeding to remove unattached epithelial cells, red blood and immune cells. The media change was also used to isolate uNK cells (see below). Isolated EnSCs were cultured in 10% DMEM/F12 media: DMEM/F12 with phenol red (Gibco, Fisher Scientific, Loughborough, UK), 10% dextran-coated charcoal-treated foetal bovine serum (DCC-FBS; see below), 10 µM L-glutamine (Gibco), $1 \times$ Antibiotic Antimycotic (Gibco), 1 nM β-estradiol and 2 µg/ml recombinant human insulin (Sigma-Aldrich, Poole, UK)). EnSCs were maintained at 37°C in a 5% $CO_2$ humidified environment in 10% DMEM/F12. For decidualization experiments, confluent EnSC monolayers were down-regulated in 2% DMEM/F12 media (DMEM/F12 media without phenol red, 2% DCC-FBS, $1 \times$ Antibiotic Antimycotic and 10 µM L-glutamine) overnight before treatment with 0.5 mM 8-bromo-cAMP (Sigma-Aldrich, Poole, UK) and 1 µM medroxyprogesterone acetate (MPA) (both Sigma-Aldrich). For routine passage, cultured EnSCs were lifted by 3 min incubation with 0.25% trypsin at 37°C and gentle agitation. Unless stated otherwise, cells were discarded after passage 3. Cells were treated with specified compounds in accordance with the experimental parameters indicated. Time-matched vehicle controls ensure effects are mediated by independent variables. For preparation of DCC-FBS, fetal bovine serum was stripped of non-polar material including steroid hormones by pre-treatment with dextran-coated charcoal. Serum was treated by suspending 1.25 g of activated charcoal and 125 mg of dextran in 500 ml serum. The solution was incubated at 56°C for 2 hr with shaking at 30 min intervals. Charcoal was removed by centrifugation ($1800 \times g$, 30 min, RT), and the supernatant sterile-filtered through 40 µm Stericup filter units (Merck Millipore, Billerica, MA, USA).

## Isolation and culture of uNK cells

Supernatant from freshly digested EnSC cultures was collected 6–18 hr post-seeding and centrifuged (200 × $g$, 5 min, RT) and red blood cells were excluded through Ficoll-Paque density gradient centrifugation. Magnetic activated cell separation (MACS; Miltenyi Biotec, Bergisch Gladbach, Germany) was performed using CD56 antigen for uNK cell isolation. Cells were re-suspended in separation buffer (0.5% BSA in PBS, sterile filtered), and counted with a LUNA BF automated cell counter (Logos Biosystems, South Korea). Cell suspensions were adjusted to a maximum density of $1 \times 10^6$ cells per 100 μl and incubated for 30 min at 4°C with 1:20 phycoerythrin (PE)-conjugated anti-CD56 antibody (Bio-Legend, San Diego, CA, USA). Cells were washed free of unbound antibody by addition of 1 ml separation buffer and centrifuged (280 g, 5 min, RT). Pellets were re-suspended in 80 μl separation buffer and 20 μl anti-PE-microbeads (Bio-Legend) per $1 \times 10^7$ cells and incubated for 30 min at 4°C. Cells were again washed by addition of 1 ml separation buffer and collected by centrifugation (280 × g, 5 min, RT). Cells were then re-suspended in 500 μl separation buffer and labelled microbeads were isolated by passing through magnetic separation (MS) columns within a magnetic field, as per manufacturer's instructions (Miltenyi Biotec). Unlabeled cells passed through the column to waste, whilst magnetically labelled CD56$^+$ cells were retained. The columns were removed from the magnetic field and CD56$^+$-cells were flushed out with 1 ml separation buffer. For increased purity, magnetic separation was repeated on the positive fraction using a fresh MS column. CD56$^+$ cells were cultured in suspension in RPMI media (Sigma-Aldrich) supplemented with 10% DCC-FBS, 1 × Antibiotic Antimycotic and 2 ng/ml IL-15 (Sigma-Aldrich) to aid uNK cell maturation. To increase yield, uNK harvests from 3 to 5 patients were pooled, and never used beyond 7 days in culture. For co-culture experiments, uNK cells were pelleted and re-suspended in 2% DMEM/F12 without IL-15 and co-cultured at half the EnSC seeding density. Inhibitors were pre-incubated with EnSCs for 1 hr prior to and during co-cultures as follows: Z-VAD-FMK (apoptosis inhibitor) (10 μM), 3,4-DCI (25 μM) (granzyme activity inhibitor), anti-IL-15 inhibitory antibody (1 μg/ml, MAB247, Bio-Techne, Abingdon, UK) and inhibitory anti-NKG2D antibody (10 μg/ml, MAB139, Bio-Techne).

## siRNA transfection

Primary EnSCs were transfected using jetPRIME Polyplus transfection kit (VWR International, Lutterworth, UK) exactly as per manufacturer's instructions. Confluent EnSC monolayers in 24-well plates were downregulated for 18–24 hr in supplemented 2% DMEM/F12 media and transfected with 50 nM siGenome SMARTpool siRNA targeting IL-15 or FOXO1 (Dharmacon, GE Healthcare), or 50 nM MISSION esiRNA targeting *CXCL8* (IL-8). Cells were also transfected in parallel with 50 nM non-targeting (NT) scrambled siRNA for negative control. Media was changed 18 hr post transfection and cells decidualized with cAMP and MPA as required. Knock-down was confirmed by RTq-PCR, Western blot analysis and/or ELISA.

## In vitro colony-forming unit (CFU) assay

CFU assays were performed as described (*Lucas et al., 2016b*; *Murakami et al., 2013*), following the treatment regimen described for individual experiments. Briefly, 500 EnSCs were seeded per well (50 cell/cm$^2$) into 10 μg/ml fibronectin-coated 6-well plates and cultured in 10% DMEM/F12 containing 10 ng/ml basic fibroblast growth factor for 12 days with a 50% media change on day 7. Careful attention was made not disturb cells during first 24–48 hr of culture. Cultures were examined periodically to confirm colonies arose from single cells. At termination, wells were washed free of media with PBS and fixed in 4% formalin for 10 min at RT before colonies were stained with hematoxylin for 3 min. Colonies were visualized on an EVOS AUTO microscope (ThermoFisher Scientific) with a 4x objective lens using scan and stitch modalities, with post-editing in ImageJ image analysis software to smooth joins. Colonies of more than 50 cells were counted. Cloning efficiency (%) was calculated as the number of colonies formed/number of cells seeded ×100.

## Real-time quantitative (RTq)-PCR

Total RNA was isolated using STAT-60 (AMS Biotechnology, Oxford, UK), reverse transcribed from 1 μg RNA using Quantitect Reverse Transcription kit (Qiagen, Hilden, Germany), including genomic DNA digestion, and assayed in triplicate by real-time PCR using Power SYBR Green Master Mix (Fisher Scientific, Loughborough, UK), all according to manufacturer's instructions. Thermal cycling

and fluorescence detection was performed on a 7500 Real-time PCR System (Applied Biosystems, Paisley, UK). Relative gene expression levels were calculated using the Pfaffl method (*Pfaffl, 2001*) and expressed as arbitrary units. *RPL19* (*L19*; coding ribosomal protein L19) was used as a reference gene. Amplification specificity of primers was confirmed by melting curve analysis and agarose gel electrophoresis, and efficiency calculated from standard curves.

Primer sequences used were as follows: *PRL* sense 5'-AAG CTG TAG AGA TTG AGG AGC AAA C-3', *PRL* antisense 5'-TCA GGA TGA ACC TGG CTG ACT A-3', *IGFBP1* sense 5'-CGA AGG CTC TCC ATG TCA CCA-3', *IGFBP1* antisense 5'-TGT CTC CTG TGC CTT GGC TAA AC-3'; *NOX4* sense 5'-AAT TTA GAT ACC CAC CCT C-3', *NOX4* antisense 5'-TCT GTG GAA AAT TAG CTT GG-3' *IL-8* sense 5'–CCA GGA AAA ACT GGG TGC AGA-3', *IL-8* antisense 5'-TTC ACT GAT CTT TGG ATA CCA CAG-3', *IL-15* sense 5'-AGC AAT GTT CCA TCA TGT TC-3', *IL-15* antisense 5'-ATA CGA TCT TGT ATG GGC TG-3', *FOXO1* sense 5'-TGG ACA TGC TCA GCA GAC ATC-3', *FOXO1* antisense 5'-TTG GGT CAG GCG GTT CA-3' and *L19* sense 5'-GCG GAA GGG TAC AGC CAA-3', *L19* antisense 5'-GCA GCC GGC GCA AA-3'.

## Enzyme-linked immunosorbent assay

Cultured EnSCs in 2% DCC/DMEM-F12 were treated in accordance with experimental schedules. Conditioned media was harvested from cells every 2 days and cleared of debris via centrifugation (16000 g, 10 min 4°C). Cells were harvested at this stage in ice-cold RIPA buffer containing cOmplete Mini protease inhibitors (Roche, Basel, Switzerland) for protein quantitation via Bradford assay (*Bradford, 1976*). ELISAs were performed exactly as per manufacturer's instructions (DuoSet ELISA kits for IL-8 (D8000C), IL-15 (D1500), GROα (DY275) and IL-6 (DY206); Bio-Techne, Abingdon, UK). Standard curves were fitted to a 4-parameter logistic fit curve in GraphPad Prism software and sample concentrations interpolated from these graphs. Final values were calculated per mg protein determined from cell harvest.

## Senescence-associated-β-galatosidase (SAβG) staining

SAβG staining was performed on confluent EnSC in 24-well plates using Senescence β-Galactosidase Staining Kit (9860; Cell Signalling Technology, MA, USA) according to the manufacturer's instruction. Detection of β-galactosidase at pH 6.0 is a known phenotype of senescent cells, and is absent from pre-senescent, proliferative, quiescent or immortal cells (*Dimri et al., 1995*). Briefly, cells were fixed in 1x fixative solution (formaldehyde-glutaraldehyde mix) for 10 min at RT and stained overnight at 37°C in air in a non-humidified incubator. Plates were sealed to prevent evaporation. Staining solutions were aspirated the next day and cells overlaid with 70% (v:v) glycerol in $H_2O$ and stored at 4°C. Staining was visualized by light microscopy and images taken.

## Quantitation of saβg activity

SAβG accumulation in cell and whole tissue lysates was quantified using the 96-Well Cellular Senescence Activity Assay kit (CBA-231, Cell Biolabs Inc; CA, USA), according to the manufacturer's instructions with minor modification. The assay quantifies β-Galactosidase activity under acidic conditions by measuring a fluorescent cleavage product. For cell culture analysis, EnSCs were seeded into 96-well plates at a density of 25,000 cells/well and treated as required. At culture termination, cells were washed in PBS and lysed in 50 µl ice-cold lysis buffer containing protease inhibitors (cOmplete mini tablets) and incubated on ice for 15 min. Lysates were assayed immediately or plates were sealed and stored at −20°C. In transfection experiments, cells were seeded in 24-well plates at a density of 100,000 cells/well and grown for 2 days before transfection. At culture termination, cells were lysed in 120 µl SAβG lysis buffer (with protease inhibitors). For whole tissue analysis, tissue was washed free of excess blood by transfer through ice-cold PBS before lysis in 500 µl SAβG cell lysis buffer (with protease inhibitors) using a hand-held POLYTRON homogeniser (Kinematica, Lucerne, Switzerland). Tissue debris and un-lysed cells were cleared via centrifugation (10,000 × *g*, 2 min, 4°C), and the supernatant quantified for SAβG as described above. The assay was performed in non-treated 96-well plates, where 30 µl cell lysate was incubated with 30 µl reaction mixture [composition: 2 × reaction buffer, 5% (v:v) SAβG substrate and 0.1 nM β-mercaptoethanol] for 1 hr at 37°C in air in a non-humidified incubator. Cell lysis buffer alone was used as control. The assay was terminated by transfer of 50 µl reaction mix to black-walled, black-bottomed 96-well plates and addition

of 200 µl stop solution. Fluorescence was determined on a PheraStar fluorescent plate reader (BMG Labtech, Ortenberg, Germany) at 360 nm (excitation) and 465 nm (emission). Fluorescent output was used as a direct measure of SAβG activity. Assays were normalized by consistent cell seeding (EnSCs) or total protein content (whole tissue) as determined by the Bradford method (*Bradford, 1976*).

## Western blot analysis

Cells cultured and treated in 6-well plates were washed twice in PBS and lysed by addition of 200 µl ice-cold high-salt lysis buffer (50 mM Tris-HCl, pH 8.0, 500 mM NaCl, 5 mM EDTA, 1% (v:v) NP-40 and cOmplete mini protease inhibitors (Roche, Basel, Switzerland)). Lysate was harvested by scraping and incubated on ice for 10 min. Cell debris was cleared through centrifugation (10,000 × *g*, 2 min, 4°C) and supernatant collected. Protein content was determined by Bradford assay (*Bradford, 1976*) and adjusted to 1 mg/ml with lysis buffer. Samples were prepared in 25% (v:v) NuPage LDS 4 × sample buffer (Fisher Scientific) and 100 nM DTT, and heated at 100°C for 5 min before separation via Western blotting. Whole frozen endometrial biopsies were washed free of excess blood by transfer through clean ice-cold PBS, before homogenisation in 500 µl ice-cold high-salt lysis buffer using a hand-held POLYTRON homogeniser. Tissue debris and un-lysed cells were cleared via centrifugation (10,000 x *g*, 2 min, 4°C) and the protein content of supernatant determined and adjusted as above.

Lysates (25 µg per lane) were loaded into 12% mini poly-acrylamide gels using the Invitrogen XCell Surelock (Carlsbad, CA, USA) Western blotting system and separated by standard SDS-PAGE electrophoresis. Proteins were transferred onto 0.45 µm nitrocellulose (GE Healthcare, Amersham, UK) and blocked in 5% powdered milk (w:v) in TBS-T (50 mM Tris, 150 mM NaCl, 0.5% (v:v) Tween-20, pH 7.4) for 1 hr. Membranes were probed with antibodies targeting Lamin B1 (1:1000; ab16048, Abcam, Cambridge, UK), HMGB2 (1:500; ab67282, Abcam), tumour suppressor genes p16[Ink4a] (1:1500; ab108349, Abcam) and p53 (1:3000; Agilent, Santa Clara, USA: M7001), and the senescence-associated heterochromatin markers Histone H1 (1:2000; Abcam: ab125027), MacroH2A (1:5000; ab183041, Abcam) H3k9me3 (1:1000; ab8898, Abcam), FOXO1 (1:1000; 2880S, Cell Signaling Technologies, Denvers, M.A, USA) and β-actin, (1:100,000; A5441, Sigma-Aldrich). Antibodies were incubated overnight in TBS-T at 4°C. Blots were washed clear of unbound antibody in TBS-T before addition of anti-rabbit-HRP or anti-mouse-HRP conjugated secondary antibody (1:1000 in TBS-T; Agilent) for 1 hr at RT. Unbound antibody was washed in TBS-T and immune-reactive bands visualized with ECL reagent (GE Healthcare, Amersham, UK) and standard auto-radiography techniques. The density of individual bands was determined using Genetools gel analysis software (Syngene, Cambridge, U.K.) and plotted using GraphPad Prism software v6 (GraphPad Software Inc. CA, USA).

## Immunocytochemistry

Cytospin preparations from 100,000 isolated uNK cells were prepared using EZ Single Cytofunnels (ThermoFisher Scientific) and Surgipath X-tra Adhesive microslides (Leica Biosystems, Wetzlar, Germany). Cells were fixed in 10% formalin for 10 min, washed in TBS-T and probed with anti-CD56 antibody overnight at 4°C (1:250; clone 123C3, Agilent). CD56[+] cells were identified using the Novolink polymer detection system exactly as per manufacturer's instructions (Leica Biosystems). Cells were also stained with H&E. Primary antibody negative slides were used as controls. For immunofluorescence analysis, treated EnSCs seeded (100,000 cells per dish) in collagen-coated 35 mm glass-bottomed culture dishes (MatTek Corporation, Ashland, MA, USA) were fixed in 4% paraformaldehyde for 10 min, RT, and permeabilized in 0.5% Triton X-100 for 10 min, RT. Cells were probed with antibodies targeting H3K9me3 (1:500; Abcam), MacroH2A (1:2000; Abcam), Histone H1 (1:500; Abcam) and Lamin B1 (1:500; Abcam) overnight at 4°C followed by TBS-T washes before incubation with Alexa Fluor 488 anti-rabbit secondary antibody for 1 hr at RT (1:1000; Fisher Scientific). Cells were counterstained with DAPI (ProLong Gold Antifade Mountant with DAPI (ThermoFisher Scientific) and washed with TBS-T and imaged on an inverted confocal microscope (Axiovert 200 M) equipped with an LSM 510 META confocal scanner (Carl Zeiss, UK). Confocal images were captured with a 488 nm wavelength scanning laser and emitted light recorded through a band-pass filter (505–530 nm). Confocal settings were matched between samples to permit comparison of relative fluorescence. For

quantification of p16$^+$ cells, EnSCs in 6-well plates were fixed and permeabilized as above, then probed with anti-p16$^{INK4}$ overnight at 4°C (1:400; Abcam) followed by TBS-T washes then Alexa Fluor 594 secondary antibodies for 1 hr at RT (1:1000; Fisher Scientific). Cells were DAPI stained and images captured on an EVOS FL Auto fluorescence microscope (Life Technologies, Paisley, UK). The number of fluorescent cells within individual p16$^+$ and DAPI images were counted using ImageJ image analysis software. p16$^+$ cells were expressed as a percentage of total (DAPI-stained) cells.

## Immunohistochemistry

Endometrial biopsies were fixed overnight in 10% neutral buffered formalin at 4°C and wax embedded in Surgipath Formula 'R' paraffin using the Shandon Excelsior ES Tissue processor (Thermo-Fisher). Tissues were sliced into 3 µM sections on a microtome and adhered to coverslips by overnight incubation at 60°C. Deparaffinization, antigen retrieval (sodium citrate buffer; 10 mM sodium citrate, 0.05% Tween-20, pH 6), antibody staining, hematoxylin counter stain and DAB colour development were fully automated in a Leica BondMax autostainer (Leica BioSystems). Tissue sections were stained for CD56 (a uNK-specific cell surface antigen) using a 1:200 dilution of concentrated CD56 antibody (NCL-L-CD56-504, Novocastra, Leica BioSystems), or p16 using a 1:5 dilution of p16$^{Ink4a}$ antibody (CINtec clone E6H4, Roche, Basel, Switzerland) as required. Stained slides were de-hydrated, cleared and cover-slipped in a Tissue-Tek Prisma Automated Slide Stainer, model 6134 (Sakura Flinetek Inc. CA, USA) using DPX coverslip mountant. Bright-field images were obtained on a Mirax Midi slide scanner using a 20x objective lens and opened in Panoramic Viewer v1.15.4 (3DHISTECH Ltd, Budapest, Hungary) for analysis. To avoid inconsistencies that reflect reduced uNK cell densities at greater endometrial depths, CD56 positive cells were quantified in compartments directly underlying the luminal epithelium. Here, three randomly selected areas of interest for each biopsy were identified and captured within Panoramic Viewer before analysis in ImageJ image analysis software (Rasband W.S. ImageJ, National Institutes of Health). Both luminal and glandular epithelial cells were removed manually before color deconvolution into constituent brown (CD56$^+$ staining) and blue (hematoxylin staining – stromal cells) (*Ruifrok and Johnston, 2001*). The area of positive staining above a manually determined background threshold was used to quantify staining intensity. The uNK percentage from these data were calculated as CD56/stromal cells $\times$ 100 (*Drury et al., 2013*), and averaged from three images. For uNK: stromal cell correlations, 4 regions of interest strictly within only the stromal cell compartment were selected. To avoid bias regions were selected whilst blinded to the corresponding uNK stains. P16$^+$ cells were quantified separately in luminal epithelium, glandular epithelium and stromal compartments. Individual compartments were isolated manually and colors deconvolved into constituent brown (p16$^+$cells) and blue (hematoxylin staining) and staining quantified as described above. To obtain confidence intervals, p16 or uNK scores at each day of the cycle were fitted into a beta distribution with the fitdistr function from the MASS package version 7.3–45 (*Venables and Ripley, 1997*) in R version 3.2.1 (*Team, 2015*). The resulting percentile values were smoothed with spline interpolation using the spline function of the basic stats package of R and converted into a heatmap using the Multi Experiment Viewer (MeV) version 4.9.0 (*Saeed et al., 2003*).

## xCELLigence

Cell viability was assessed in real-time using an xCELLigence Real Time Cell Analysis (RTCA) DP instrument (ACEA Biosciences Inc, San Diego, CA, USA). EnSCs were seeded in E-16 xCELLigence plates at a density of 20,000 cells/well and cultured for 24 hr before overnight down-regulation in 2% DCC. Cells were decidualized with cAMP or MPA for 8 days as per standard protocols or left undifferentiated (vehicle control/day 0). On the day of assay, uNK cells were collected from suspension culture by centrifugation (1200 rpm, 5 min, RT) and re-suspended in 2% DCC and 10,000 uNK cells/wells were added to EnSCs and incubated for 30 min prior to assay to allow cells to settle. The xCELLigence was referenced to an impedance of 0, and changes measured every 15 min for 24 hr. For experiments requiring inhibitors, cells were treated with 10 µM Z-FAD-FMK, a broad-spectrum caspase inhibitor or 25 µM 3,4-Dichloroisocoumarin (3,4-DCI), a granzyme inhibitor, for 24 hr prior to assay and co-cultured with uNK cells in the continued presence of the inhibitors.

## Statistical analysis

GraphPad Prism v6 (GraphPad Software Inc.) was used for statistical analyses. Data were checked for normal distribution using Kolmogorov-Smirnov test. Unpaired or paired t-test was performed, as appropriate, to determine statistical significance between groups for normally distributed data. For comparing more than two groups, the data were analyzed using ANOVA, followed by Tukey's or Dunnett's post-hoc test for multiple comparisons. Pearson's correlation coefficient was used to assess the strength of the linear relationship between data sets. $p < 0.05$ was considered significant.

## Acknowledgement

We are grateful to all the women who participated in this research. We thank Gnyaneshwari Patel, Anatoly Shmygol, and Jesús Gil for advice and technical assistance. This work was supported by funds from the Tommy's National Miscarriage Research Centre and the Biomedical Research Unit in Reproductive Health.

## Additional information

### Funding

| Funder | Grant reference number | Author |
|---|---|---|
| Tommy's National Miscarriage Research Centre | RMRRH0035 | Pavle Vrljicak<br>Shreeya Tewary<br>Emma Lucas<br>Siobhan Quenby<br>Jan Joris Brosens |
| BioMedical Research Unit in Reproductive Health, University Hospitals Coventry and Warwickshire | RMRRH0019 | Paul J Brighton<br>Yojiro Maruyama<br>Katherine Fishwick<br>Risa Fujihara<br>Joanne Muter<br>Taihei Yamada<br>Raffaella Lucciola |

The funders had no role in study design, data collection and interpretation, or the decision to submit the work for publication.

### Author contributions

Paul J Brighton, Data curation, Formal analysis, Supervision, Investigation, Methodology, Writing—original draft, Writing—review and editing; Yojiro Maruyama, Joanne Muter, Laura Woods, Data curation, Formal analysis, Investigation, Methodology; Katherine Fishwick, Yie Hou Lee, Formal analysis, Investigation, Methodology; Pavle Vrljicak, Data curation, Formal analysis; Shreeya Tewary, Investigation, Methodology; Risa Fujihara, Taihei Yamada, Raffaella Lucciola, Methodology; Emma S Lucas, Data curation, Formal analysis, Investigation, Methodology, Writing—original draft; Satoru Takeda, Siobhan Quenby, Supervision, Funding acquisition; Sascha Ott, Data curation, Formal analysis, Supervision; Myriam Hemberger, Data curation, Formal analysis, Supervision, Methodology; Jan Joris Brosens, Conceptualization, Data curation, Formal analysis, Supervision, Funding acquisition, Validation, Investigation, Writing—original draft, Project administration, Writing—review and editing

### Author ORCIDs

Yie Hou Lee 🆔 https://orcid.org/0000-0003-2723-2023
Jan Joris Brosens 🆔 http://orcid.org/0000-0003-0116-9329

### Ethics

Human subjects: The study was approved by the NHS National Research Ethics - Hammersmith and Queen Charlotte's & Chelsea Research Ethics Committee (1997/5065). Subjects were recruited from the Implantation Clinic, a dedicated research clinic at University Hospitals Coventry and Warwickshire National Health Service Trust. Written informed consent was obtained from all participants in accordance with the guidelines in The Declaration of Helsinki 2000.

Decision letter and Author response
Decision letter https://doi.org/10.7554/eLife.31274.023
Author response https://doi.org/10.7554/eLife.31274.024

## Additional files

### Supplementary files

• Supplementary file 1. Patient demographics. Data are related to primary cultures or individual figures as indicated. Primary cultures refer to all biopsies from which EnSCs were isolated and propagated in culture. *Values are presented as mean ±SD. Values are presented as median (range). N/A: not applicable.
DOI: https://doi.org/10.7554/eLife.31274.020

• Transparent reporting form
DOI: https://doi.org/10.7554/eLife.31274.021

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
