## [Decision Letter]

Thank you for submitting your article "Clearance of senescent decidual cells by uterine natural killer cells in cycling human endometrium" for consideration by *eLife*. Your article has been reviewed by three peer reviewers, and the evaluation has been overseen by a Reviewing Editor and Mark McCarthy as the Senior Editor. The following individual involved in review of your submission has agreed to reveal his identity: Gunter Wagner (Reviewer #1).

The reviewers have discussed the reviews with one another and the Reviewing Editor has drafted this decision to help you prepare a revised submission.

Summary:

In this study Brosens and collaborators present a novel hypothesis about the role of cell senescence during the implantation phase of pregnancy. The thesis is that a sub-population of endometrial stromal fibroblasts undergo cellular senescence and in the course of that produce a cocktail of inflammatory signals that leads to the inflammatory activation necessary for implantation. Later these cells get removed leading to the remodeling of the extracellular matrix encouraging implantation. This model is supported by experiments with primary endometrial cell culture and a large amount of clinical data.

Essential revisions:

1) Co localization of some of the markers such as p16 and SABG should be conducted to ensure this sole population of cells.

2) When conducting withdrawal of cAMP MPA they see an increase in the senescent markers they should address if this is maintenance of senescence of cell death due to hormonal support.

3) There are multiple seemingly independent approaches that the authors piece together in the final analysis which would be greatly improved by a schematic diagram of what they think is going on based on the data presented.

4) Importantly, the clinical data about the human subjects is incomplete. Did the samples come from women with no abnormalities and with proven fertility? Gynecologic disorders? Ovulatory disorders? Endocrine disorders? Recurrent miscarriage? Recurrent implantation failure? Unexplained infertility? How do the authors know whether their observations are unique to women with abnormal reproductive outcomes?

---

## [Author Response]

Essential revisions:1) Co localization of some of the markers such as p16 and SABG should be conducted to ensure this sole population of cells.

This is a very good suggestion; in fact, we did attempt co-localisation studies for p16 and SABG in primary cultures. However, these assays use different, non-compatible fixation methods, which precluded further analysis. Specifically, SABG staining requires glutaraldehyde, methanol and formaldehyde fixation, which does not permit p16 immunocytochemistry (ICC). Conversely, fixation of cells for ICC in 4% paraformaldehyde and 0.2% Triton-X precluded SABG staining. Other fixation methods were equally unsuccessful.

2) When conducting withdrawal of cAMP MPA they see an increase in the senescent markers they should address if this is maintenance of senescence of cell death due to hormonal support.

We wish to point out that senescence markers do not rise but remain stable upon withdrawal of 8-br-cAMP and MPA from primary cultures decidualized for 8 days (Figure 3). We reported previously a modest increase in cell death (from 3.5% to 7%) in response to withdrawal of 8-br-cAMP and MPA from decidual cultures (Labied et al., 2006). The data infer that withdrawal of deciduogenic signals is insufficient to elicit significant cell death of senescent decidual cells.

3) There are multiple seemingly independent approaches that the authors piece together in the final analysis which would be greatly improved by a schematic diagram of what they think is going on based on the data presented.

Thank you for this great suggestion. A schematic diagram is included in the revised manuscript (new Figure 7).

4) Importantly, the clinical data about the human subjects is incomplete. Did the samples come from women with no abnormalities and with proven fertility? Gynecologic disorders? Ovulatory disorders? Endocrine disorders? Recurrent miscarriage? Recurrent implantation failure? Unexplained infertility? How do the authors know whether their observations are unique to women with abnormal reproductive outcomes?

In subsection “Patient recruitment and sample collection”, we expanded on the clinical presentation of participating patients as follows: ‘The study cohort consisted of 942 women suffering recurrent miscarriage (defined as 3 or more consecutive miscarriages), 349 patients with recurrent implantation failure (define as 3 or more consecutive IVF failures), 139 women presenting with recurrent miscarriage following IVF treatment, and 681 other subjects, mostly women awaiting IVF treatment for a variety of reproductive disorders’.

We wish to point out that the aim of the centile graphs is to highlight temporal and inter-patient variation, whereas cycle-to-cycle variation is illustrated in Figure 6 and Figure 6—figure supplement 1. Importantly, there are no unequivocal criteria to define ‘normal’ endometrium. As outlined in the Discussion section, implantation failure and miscarriage do not preclude a future successful pregnancy. Indeed, cumulative live birth rates are high for patients suffering recurrent miscarriage or repeated IVF failure. Parous women are often considered ‘normal fertile controls’, although this assumes that a successful pregnancy has no discernible impact on subsequent endometrial responses, which is contentious. Further, a successful pregnancy does not preclude secondary recurrent miscarriage or implantation failure. Our findings do suggest that lack of decidual senescence and/or uNK hyperactivation may contribute to implantation failure whereas excessive decidual senescence and/or uNK deficiency are predicted to predispose for miscarriage. These hypotheses can only be tested rigorously in adequately powered, prospective clinicalstudies that link analysis of timed endometrial biopsies to subsequent pregnancy outcome. Such studies are planned.